# Workers’ Exposure Assessment during the Production of Graphene Nanoplatelets in R&D Laboratory

**DOI:** 10.3390/nano10081520

**Published:** 2020-08-03

**Authors:** Irene Bellagamba, Fabio Boccuni, Riccardo Ferrante, Francesca Tombolini, Fabrizio Marra, Maria Sabrina Sarto, Sergio Iavicoli

**Affiliations:** 1Research Center for Nanotechnology Applied to Engineering (CNIS), Sapienza University of Rome, I-00185 Rome, Italy; fabrizio.marra@uniroma1.it (F.M.); mariasabrina.sarto@uniroma1.it (M.S.S.); 2Department of Astronautical, Electrical and Energy Engineering (DIAEE), Sapienza University of Rome, I-00184 Rome, Italy; 3Italian Workers’ Compensation Authority—Department of Occupational and Environmental Medicine, Epidemiology and Hygiene, Via Fontana Candida 1, I-00078 Rome, Italy; f.boccuni@inail.it (F.B.); ri.ferrante@inail.it (R.F.); f.tombolini@inail.it (F.T.); s.iavicoli@inail.it (S.I.)

**Keywords:** nanotechnologies, graphene nanoplatelets, 2D nanostructures, occupational safety and health, exposure assessment, risk mitigation

## Abstract

Widespread production and use of engineered nanomaterials in industrial and research settings raise concerns about their health impact in the workplace. In the last years, graphene-based nanomaterials have gained particular interest in many application fields. Among them, graphene nanoplatelets (GNPs) showed superior electrical, optical and thermal properties, low-cost and availability. Few and conflicting results have been reported about toxicity and potential effects on workers’ health, during the production and handling of these nanostructures. Due to this lack of knowledge, systematic approaches are needed to assess risks and quantify workers’ exposure to GNPs. This work applies a multi-metric approach to assess workers’ exposure during the production of GNPs, based on the Organization for Economic Cooperation and Development (OECD) methodology by integrating real-time measurements and personal sampling. In particular, we analyzed the particle number concentration, the average diameter and the lung deposited surface area of airborne nanoparticles during the production process conducted by thermal exfoliation in two different ways, compared to the background. These results have been integrated by electron microscopic and spectroscopic analysis on the filters sampled by personal impactors. The study identifies the process phases potentially at risk for workers and reports quantitative information about the parameters that may influence the exposure in order to propose recommendations for a safer design of GNPs production process.

## 1. Introduction

In recent years nanomaterials have undergone a rapid development in a wide range of sectors, due to their extraordinary properties that allowed them to gain multidisciplinary relevance. In parallel with the high diffusion, production and use of different nanomaterials, the research community started to yield interest in their potential impact for workers involved in industrial and small-scale manufacturing processes.

Several scientific studies confirm that a nanomaterial can be more hazardous than the corresponding material in the bulk form [1] due to the large effective surface area per mass unit [2,3].

A lot of scientific studies regarding in vitro and in vivo toxicity assessment of carbon-based nanomaterials such as single walled carbon nanotubes (SWCNTs) [4,5] or multiwalled carbon nanotubes (MWCNTs) [6,7,8] demonstrate that they cause germ cell mutagenicity, specific organ toxicity (SWCNTs, MWCNTs), eye damage and carcinogenicity (MWCNTs) after repeated exposure.

In the last years, many of the R&D facilities that previously worked on carbon nanotubes now changed to graphene-based nanomaterials such as graphene (G), few layer graphene (FLG), graphene oxide (GO), reduced graphene oxide (rGO) and graphene nanoplatelets (GNPs).

In particular, recent scientific breakthroughs have demonstrated that GNPs, in addition to their significant physiochemical properties, show great antimicrobial effects [9], without exerting relevant cytotoxic damages on human cell lines [10]. However, based on the available literature data, owing to the high uncertainty and the numerous conflicting outcomes about the hazardous properties of graphene nanomaterials [11,12,13] and their possible effects on humans’ health [14], specific occupational exposure limits are not yet available. In addition, only few studies are focused on the assessment of workers’ potential exposure to GNPs considering the real workplace settings and the research labs facilities [15,16,17]. Therefore, to deal with this lack of knowledge, it is necessary to put in place suitable and effective methodologies for measuring the occupational exposure of workers involved in the production and handling processes of nanomaterials [18].

Based on the harmonized and tiered approach proposed by the Organization for Economic Cooperation and Development (OECD) [19] the Italian Workers’ Compensation Authority (INAIL) has developed a multi-metric methodology to measure the potential exposure to engineered nanomaterials in the workplace [20].

In this framework, the present study aims to evaluate levels of workers’ exposure against nanoparticle emission during GNPs production, with the scope of proposing specific and suitable risk mitigation strategies, considering two different routes of manufacturing and their single work phases. Graphene nanoplatelets were produced by thermal exfoliation of graphite intercalated compounds (GICs), applying the methodology developed at Sapienza University of Rome [21,22]. The exposure to GNPs has been assessed by integrating different instruments and methodologies, including real-time measurements to detect the nanomaterials size, number concentration and surface area and personal sampling to conduct off-line analysis with the aim of identifying and characterizing the airborne nanomaterials in the workplace.

## 2. Materials and Methods 

### 2.1. Graphene Nanoplatelets

GNPs are bidimensional carbon nanostructures consisting of stacks of graphene sheets, having a thickness in the nanometer range and lateral dimensions in the micrometer range. GNPs are non-metal materials, but they are electrically conducting nanostructures usually known for their interesting electrical, electromagnetic and piezoelectric properties [23]. Therefore, these nanostructures represent a good candidate to replace many carbon-based conductive nanomaterials such as carbon nanotubes, carbon nanofibers, carbon dots and fullerenes, for many applications, due to the low manufacturing cost and easy production scalability. Graphite intercalated compounds (GICs) are made up of parallel layers plane of graphite facets with an expansion agent (or intercalant), they have an irregular spherical shape with a mean lateral size of 350 μm (see Figure 1a). A scanning electron microscopy (FE-SEM, Zeiss Auriga, Oberkochen, Germany) image of the expanded graphite obtained after the thermal shock of GICs is shown in Figure 1b: it is also called “worm-like” expanded graphite (WEG) because of its particular shape. In order to assess the morphological properties of GNPs, we performed SEM and scanning transmission electron microscopy (STEM) analysis on the raw material (Figure 1c–f) by using a high-resolution field emission-scanning electron microscope (FE-SEM, Zeiss Auriga, Oberkochen, Germany). 

As shown in the SEM images reported in Figure 1c,d GNPs had lateral dimensions in the micrometer-range (between 1 and 5 μm) and they were characterized by an irregular geometry and by sharp edges and wedges.

In order to evaluate the thickness of the produced GNPs, we assessed an atomic force microscopy (AFM) analysis, using a Bruker-Veeco Dimension Icon AFM (Billerica, MA, USA) available in the Nanotechnology and Nanoscience Laboratory of Sapienza University (SNN-Lab). Figure 2a shows an AFM image of a single GNP with stacked graphene sheets. The measured profile of this structure (Figure 2b) shows a thickness varying from 7 to 18 nm [24]. The GNP lateral dimensions were of around 1 µm^2^ and the average surface area was of 1.5 μm.

### 2.2. GNPs Production Process

The GNPs production process [21] is summarized in the following steps (see Figure 3):Sample preparation: before each thermal expansion 0.7 g of intercalated expandable graphite flakes is placed inside a metal melting pot.Thermal expansion of GICs: the material is placed inside a muffle furnace. This step causes an increase in the volume of GICs up to 200 times, leading to the formation of WEGs. The thermal expansion can be conducted in two ways: GICs are placed in the muffle furnace at 1150 °C for 5 s or, alternatively, the expansion can be performed at 1050 °C for 30 s.Liquid exfoliation of WEGs: after the thermal treatment, 0.5 g of WEGs are tip sonicated in 100 mL of acetone using an ultrasonic processor in liquid phase, thus obtaining GNPs.

Both the processes with different furnace temperature can be carried out to produce GNPs, but the final material shows some differences in terms of morphology, shape, dimensions and possible applications [24]. Therefore, two different experimental campaigns have been conducted: the first GNPs production process was carried out by making a thermal expansion of the GICs in oven at 1150 °C (phase 1A), while the second one was conducted at 1050 °C (phase 2A). The thermal expansion was always followed by liquid exfoliation in acetone of WEGs as in step (iii) above, by using a probe ultrasonicator, which was conducted in the same way for both the experimental campaigns (phase 1B and 2B).

The thermal expansion process involves two workers, one that opens and closes the furnace and the other one that inserts GICs in the oven and removes WEGs from it after expansion. The liquid exfoliation involves only one worker who prepares a colloidal suspension made of WEGs and acetone for the next sonication step.

The tip sonication is performed under a chemical ventilated hood at room temperature for 40 min by using a cold jacketed beaker, maintaining the GNPs/acetone solution at a constant temperature of 5 °C, thus avoiding the possible evaporation of the solvent and the resulting risk of workers’ exposure.

The production activities were carried out by the same two workers in both the experimental campaigns. During these two operations the workers wear protective gloves, laboratory coats and full-face respirators (mod. 3M™ 6000 series) equipped with EPA filters (mod. 3M™ 6099 ABEK2 P3 series).

### 2.3. Workplace Description

GNPs are produced using equipment located in two different labs. In particular, the thermal expansion is conducted inside the laboratory no. 1 (lab 1) whereas the liquid exfoliation phase is carried out in the laboratory no. 2 (lab 2). Lab 1 has a natural ventilation system and a mechanical ventilation system (air delivery and extraction), the muffle furnace has a proper ventilation system. Lab 2 has a natural ventilation system and a mechanical ventilation system (air delivery and extraction), and it is equipped with a chemical hood.

### 2.4. Exposure Assessment Methods

During the experimental campaigns, measurements and sampling were conducted by using the following instruments for real-time and off-line analysis:Condensation particle counter (CPC mod. 3007, TSI Inc., Shoreview, MN, USA): it is an optical counter that can measure the particle number concentration (PNC (#/cm^3^)) with a time resolution of 1 s and an accuracy of ±20%. CPC can detect nanoparticles with an average size in the range between 10 nm and 1 µm.Mini diffusion size classifier (DiSCmini, mod. TESTO, Titisee-Neustadt, Germany): this portable instrument was used for measuring PNC (#/cm^3^), modal average diameter (D_avg_ (nm)) and lung deposited surface area (LDSA (µm^2^/cm^3^)) in the environment and in correspondence of the worker’s personal breathing zone (PBZ) with a time resolution of 1 s and an accuracy of ±30%.Personal impactor (mod. Sioutas, SKC Inc., Eighty Four, PA, USA): equipped with 5 different filters. It separates and collects ultrafine, fine and >2.5 µm airborne particles characterized by 5 different diameters ranges: <0.25 µm, 0.25–0.50 µm, 0.50–1.0 µm, 1.0–2.5 µm and >2.5 µm (up to 10 µm). Particles above each cut point were collected on a 25 mm aluminum filter in each appropriate stage when the Sioutas was used with a 9 L/min sample pump. Particles less than 0.25 µm cut-point of the last stage were collected on a 37 mm PTFE after-filter.Field emission scanning electron microscope (FE-SEM, Zeiss, Oberkochen, Germany) equipped with an energy dispersive X-ray spectroscopy (EDS, Oxford Instruments INCA) for off-line analysis on the collected airborne nanoparticles.

The exposure measurement method is firstly based on a comparison between the values of PNC, D_avg_ and LDSA measurements performed during the production process and the corresponding values of the background, in order to identify the nanomaterials emission sources: the difference between background and workplace concentrations can be attributed to the work with nanomaterials.

During our experimental campaigns we assessed the indoor background in two ways [25]:Far-field background (bkg_FF_): the measurements were performed in a place, within the same facility, in which no nanomaterials are produced, but simultaneously with the production process and when no production occurred.Near-field background (bkg_NF_): the measurements occurred before the production process, in the same location.

The measurements referred to phases A and B were conducted for both the experimental campaigns within two days, according to the time sheet in Table 1.

At the beginning and at the end of each measurement campaign, an instrument comparison session (parallel) has been done in order to define parameters of correlation useful to harmonize the instruments response (detailed information is reported in Appendix A).

On day 1, bkg_NF_ and bkg_FF_ were measured while there was not GNPs production: one CPC and two DiSC_mini (DM-UF3 and DM-UF4) were positioned inside each lab. On day 2, we performed the measurements during the entire GNPs production process. The locations where the instruments and personal samplers were positioned are shown in Figure 4. During the phase 1A, DM-UF3 was placed over the first worker’s lab coat for personal measurements in correspondence of his PBZ (P_A2); CPC was positioned inside lab 1 at 1.5 m from the source for area measurements (P_A1), in a location where the second worker involved in the thermal expansion phase is typically positioned in order to give assistance to the first worker; Sioutas impactor was placed over the lab coat of the second worker (P_A3). During the phase 1B, DM-UF3 and Sioutas were placed in front of the chemical hood, corresponding to the position of the worker involved in the exfoliation process (P_B2 and P_B3) and the CPC was positioned outside the chemical hood (P_B1), in a second workstation, next to the control unit of the sonicator. During the second experimental campaign a third DiSC_mini (DM-UF5) was placed inside the pre-chamber (P_AB of Figure 4) in order to evaluate the nanoparticles distribution beside the production labs. Another laboratory (lab 3), in which no nanomaterials were produced, was selected for the simultaneous bkg_FF_ measurements, by using DM-UF4. In particular, lab 3, located next to lab 2, has the same orientation, structural and ventilation properties (both natural and mechanical) as labs 1 and 2.

As from Figure 4 the laboratories were air conditioned and windows/doors were closed during the activities. Only in lab 1, at the end of the thermal expansion phase the window had opened.

The measurements performed during both the experimental campaigns are summarized in Table 1, in which the “parallel” section is referred to the measurements conducted inside each lab by using all the real-time instruments at the same time, in order to define the parameters of correlation useful to harmonize the instruments response. The “bkg_NF_-pre” and “bkg_NF_-post” sessions are referred to the background near field measurements performed before and after the production process in each lab.

## 3. Results

### 3.1. First Experimental Campaign: Phase 1A and Phase 1B

The results of the first experimental campaign, which includes the exposure measurements performed during the thermal expansion at 1150 °C (phase 1A) and during the liquid exfoliation (phase 1B), are described in the following. All the characteristic values obtained from the bkg_NF_ and bkg_FF_ measurements are resumed in Table 2.

The PNC mean values of the bkg_NF_, measured before the phase 1A by both DM-UF3 and CPC were a little lower than the bkg_FF_ (8912 #/cm^3^). On the contrary, the PNC mean values measured before the phase 1B (bkg_NF_) were higher than the mean value referred to the corresponding bkg_FF_ (9300 #/cm^3^). The bkg_NF_ referred to the phase 1A was characterized by a mean D_avg_ of 68.7 nm, higher than the the bkg_FF_ (59.0 nm). The bkg_NF_ and bkg_FF_ referred to the phase 1B were characterized by similar D_avg_ values of 45.8 nm and 49.3 nm respectively. The mean LDSA values of the bkg_NF_ and the bkg_FF_ referred to the phase 1A were similar, equal to 29 µm^2^/cm^3^. The mean LDSA value of the bkg_NF_ of phase 1B (28.8 µm^2^/cm^3^) was slightly higher than the value of the bkg_FF_ (25.1 µm^2^/cm^3^).

In order to assess the nanomaterials release during the GNPs production process, the PNC measured during the work activites should be compared with the corresponding significant values of the bkg_NF_. In particular, the PNC was statistically significant if it was higher than the corresponding significant value calculated by the sum between the mean value and three times the standard deviation [13]. In the specific case, we calculated two distinct PNC significant values related to the bkg_NF_: 11,060 #/cm^3^ for the phase 1A and 13,779 #/cm^3^ for the phase 1B.

Figure 5 shows the PNC, D_avg_ and LDSA time series referred to the measurements performed during the phase 1A inside lab 1 and inside lab 3 (bkg_FF_) compared with those performed during the phase 1B inside lab 2 and inside lab 3 (bkg_FF_). The PNC and LDSA values were normalized with respect to the maximum value recorded during the two experimental campaigns in order to compare the exposure levels between the two phases constituting each process and to understand the differences between two GNPs production methods characterized by different furnace temperature of the thermal expansion phase, as described in the following. In particular, Figure 5a,c,e shows in detail the PNC, LDSA and D_avg_ time series regarding the production phase 1A and Figure 5b,d,f shows the time series of the PNC, LDSA and D_avg_ measurements referred to phase 1B.

As reported in Figure 5a, the NPs concentration during the thermal expansion reached three distinct peaks, measured by the DM-UF3 in correspondence to the worker’s PBZ. Each peak corresponds to the opening of the oven for the removal the WEGs after each GICs expansion, at 11:41:48 a.m., 11:45:52 a.m. and 11:50:29 a.m., and they are well above the mean PNC trend of the bkg_FF_, contemporarily measured in the adjacent Lab 3. The PNC measured by the CPC during the phase 1A progressively increases, in agreement with the mean trend of the PNC recorded by the DM-UF3, but the CPC signal has a lower intensity. The LDSA time series (Figure 5c) followed the same trend of the PNC curve and they were characterized by three peaks and an average value that increased during the whole expansion process. The D_avg_ time series (Figure 5e) show that during the termal expansion the mean NPs size was about 26 nm, with a minimum value of 15 nm. Before the first expansion the average diameter was about 60 nm and it reached 15 nm in correspondence of the first opening of the furnace for the removal of WEGs, at 11:41:48 a.m. After the first expansion the D_avg_ went back to the initial mean values, but it decreased again reaching the other two minimum values in correspondence of the second (11:45:52) and the third expansion (11:50:29).

During phase 1B, as we can observe from the graph of Figure 5b, the mean PNC value measured by the DM-UF3 placed in front of the chemical hood was higher than the bkg_FF,_ however it remained quite constant during the whole liquid exfoliation phase. The same trend of the PNC can be observed from the LDSA time series (Figure 5d): the bkg_FF_ was always a little lower than the mean LDSA values measured in front of the chemical hood. The D_avg_ measured by the DM-UF3 (Figure 5f) was not subjected to high variations during the liquid exfoliation phase, ranging from 40 and 48 nm and it was lower than the D_avg_ mean value of the bkg_FF_, which increased from 50 to 56 nm.

We verified that the mean PNC trend referred to phase 1A was higher than the significant value of the bkg_NF_, whereas during the phase 1B the PNC was always lower than the corresponding bkg_NF_ significant value. The results of this experimental campaign show that the two phases of the GNPs production process were quite different in terms of the general trend over time of nanomaterials concentration, LDSA and average diameter. In particular, we might assume that during phase 1A there might be a possible release of WEGs fragments, in which there is evidence of a huge increase in the NPs concentration compared to the bkg_FF_ measurements and the bkg_NF_ significant value. On the contrary, during the phase 1B there was no evidence of GNPs release because we verified that the PNC was lower than the bkg_NF_ significant value, even if it was a little higher than the bkg_FF_. 

### 3.2. Second Experimental Campaign

The results of the second experimental campaign, which include the measurements conducted during the thermal expansion of GICs at 1050 °C (phase 2A) and the liquid exfoliation of WEGs (phase 2B) were described in the following.

The characteristic values obtained from the bkg_FF_ and bkg_NF_ measurements of PNC, D_avg_ and LDSA are summarized in Table 3.

The mean PNC values of the bkg_NF_, measured both before the phases 2A and 2B (by DM-UF3 and CPC in lab 1 and by DM-UF5 in the pre-chamber) were always higher than the corresponding bkg_FF_ levels. The average diameter of the bkg_FF_ was characterized by a mean value of 45.1 nm (phase 2A—lab 1) and 46.3 nm (phase 2B—lab 2), a little lower than the bkg_NF_ values inside the pre-chamber and in lab 1 and lab 2. Referring to both the phases, the mean LDSA values of the bkg_NF_ (measured in lab 1 and in the pre-chamber) in the range within 20.9 µm^2^/cm^3^ and 25.3 µm^2^/cm^3^, were higher than the bkg_FF_ mean values (15.9 µm^2^/cm^3^ in lab 1 and 16.6 µm^2^/cm^3^ in lab 2). 

In order to evaluate the nanomaterials release during these two phases we calculated the following two PNC significant values, referring to the bkg_NF_: 13,821 #/cm^3^ for phase 2A (lab 1) and 11,279 #/cm^3^ for phase 2B (lab 2).

The time series of PNC, D_avg_ and LDSA measurements (Figure 6a,c,e) carried out in lab 1 and inside the pre-chamber during the thermal expansion at 1050 °C (phase 2A) and contemporarily performed inside lab 3 for the bkg_FF_ measurements were compared with the time series referred to the liquid exfoliation (Figure 6b,d,f), phase 2B.

As noted above, during the phase 2B we placed the CPC near the sonicator control unit, in order to verify if there was any evidence of NPs released in correspondence of the worker who monitored the tip sonication control unit. During both the phases the DM-UF5 was placed inside the pre-chamber with the aim to evaluate the NPs distribution beside the production labs.

With reference to Figure 6a, the highest NPs concentration during phase 2A was recorded by the DM-UF3 in correspondence of the worker’s PBZ. As in the first experimental campaign, also in this case, the CPC located at 1.5 m from the working area has recorded a maximum value three times lower than the corresponding value measured by the personal instrument. As expected, the general trend of the PNC measured by the DM-UF5, located inside the pre-chamber, was below the PNC curves referred to both the DM-UF3 and the CPC measurements. The PNC measured during phase 2A by all three instruments were well above the bkg_FF_.

With reference to the personal PNC measurements (red line of Figure 6a) we can observe that in correspondence to each thermal expansion, at 10:51:08 a.m., 10:59:58 a.m. and 11:08:37 a.m., the NPs concentration significantly increased. After reaching the maximum value, at the end of each expansion process, the signal intensity slightly decreased and it started to increase again in correspondence to the next expansion. As reported in the same graph, the time series referred to the DM-UF5 measurements show that the PNC achieved the different peaks with a detection delay of about 1 min and 30 s with respect to the maximum peaks recorded by the DM-UF3, owing to the different positions of the two instruments. This delay represents the time required for the NPs to flow from one measuring point (P_A1 of Figure 4) to the other one (P_AB of Figure 4). The bkg_FF_ measurements performed in Lab 3 show that the PNC initial value remained almost constant during the whole expansion process.

As reported in Figure 6c the LDSA time series followed the same trend of the PNC curves and they are characterized by three maximum values in correspondence of each thermal expansion. During this production phase the LDSA signal was always higher than the bkg_FF_.

Before starting the first expansion, the D_avg_ (Figure 6e) measured by the DM-UF3 was about 42 nm and it reached the lowest value of 16.5 nm in correspondence to the first opening of the furnace for the removal of WEGs. At the end of the first expansion the average diameter grew up, showing a little decrease in corresponence of both the following two expansions. The D_avg_ measured by the DM-UF5 inside the pre-chamber followed the same trend of the personal monitor, reaching three minimum values of 29 nm, 31 nm and 33 nm respectively in correspondence of the three expansions, which were little higher than the personal values.

Figure 6b,d,f shows the time series of PNC, D_avg_ and LDSA referred to the liquid exfoliation (phase 2B) measurements.

As we can observe from the graph of Figure 6b, during the whole phase 2B the higher PNC signal was recorded by the DM-UF5 inside the pre-chamber, which may be caused by the external environmental pollution coming inside lab 1 from the window that was opened after phase 2A. This could have produced an increase in the NPs concentration also during the tip sonication. The DM-UF5 measurements show that at the beginning of phase 2B the PNC slightly increased until reaching a maximum value that remained almost constant until the end of the liquid exfoliation. The DM-UF3 signal intensity remained quite constant and it was not subjected to significant fluctuation around the mean value, but it was higher than the bkg_FF_. The lowest signal intensity was measured by the CPC outside the chemical hood and it was not subjected to great fluctuations around the average value, which was slightly higher than the signal simultaneously recorded in the lab 3 (bkg_FF_). The same PNC trend could be observed from the time series of the LDSA measurements (see Figure 6d).

The D_avg_ time series of Figure 6f shows that before starting the tip sonication the mean value was 55 nm inside lab 2 and 46.5 nm both inside the pre-chamber and in lab 3. During phase 2B the average diameter remained quite constant in lab 3 and in the pre-chamber, whereas it shows little fluctuations around the mean value recorded before starting the ultrasonication in lab 2 (P_B2). 

As already observed for the first experimental campaign, even then, the mean PNC trend during phase 1A was higher than the significant value of the bkg_NF_, which points out a possible nanomaterials release. Phase 2B did not reveal any nanomaterials release since the PNC mean values are always lower than the selected bkg_NF_ significant value.

### 3.3. Off-Line Analysis on Airborne Sampled Materials

As already noted from the results of the first experimental campaign, during the thermal expansion phase a possible NPs release may occur, while the liquid exfoliation phase did not seem to cause any nanomaterials release. These results will be confirmed by the SEM and EDS analysis performed on the sampled filters collected by the Sioutas personal impactor.

In order to evaluate if there is any evidence of workers’ exposure during the GNPs production, the results obtained from the real-time instruments were integrated by an off-line analysis (Feg-SEM with EDS) on the filters of the Sioutas impactor worn by the workers both during the thermal expansion and the liquid exfoliation phases. Figure 7a–c shows the results of the EDS analysis performed on the backup (<250 nm) sampled filter collected by the Sioutas during the thermal expansion. Figure 7d,e represents two magnifications of the SEM image of Figure 6a showing some isolated nanostructures similar to the produced material. Figure 7f shows the result of the SEM analysis performed on the Stage C sampled filter, in which we can observe some nanostructures referable to WEGs highlighted by the red circle, supported by airborne NPs.

The EDS anaysis was acquired along the yellow line as shown in Figure 7a in which a higher carbon signal with respect to the other signals is observed, e.g., oxigen. The green signal is referred to fluorine, which is the chemical element characterizing the teflon filter (backup filter). The low cyan signal shows the presence of gold, which was used to sputter the sample. 

The morphological investigations reported in Figure 7d–f revealed some nanostructures that were characterized by an irregular geometry and by sharp edges and wedges, with the same shape of the fragments belonging to the material produced during the thermal expansion, i.e., WEGs (see Figure 1b). In particular, the EDS analysis (Figure 7a–c) revealed the carbon signal, which could feel like the chemical composition of WEGs, made up of only carbon atoms.

From the SEM analysis conducted on the filters collected during the liquid exfoliation, there was no evidence of any material attributable to this phase, i.e., GNPs.

## 4. Discussion

The results of the real-time measurements performed during phase 1A (thermal exfoliation at 1150 °C) and during phase 2A (thermal expansion at 1050 °C) showed a clear release of NPs inside lab 1. In fact, the values of the three peaks and the average trend of PNC during the three consecutive thermal expansions were well above the bkg_FF_.

However, during phase 1A the PNC peaks were higher than those recorded during phase 2A. This can be due to the dwell time of the material inside the muffle furnace for providing the GICs thermal expansion, which lasted 30 s in phase 1A, six times more than in phase 2A (5 s). Whithin this period of 30 s, the internal ventilation system of the muffle furnace contributed to the ejection of exceeding expansion products.

From the time series of the PNC measurements it should be noted that there was a huge difference in the signal intensity measured by the real-time instruments located at different distances from the source. Figure 8 shows the residual PNC values between the personal exposure (DM-UF3) and the environmental one (CPC), during phases 1A (Figure 8a) and 2A (Figure 8b).

As we can observe from the graphs of Figure 8a,b, during both the thermal expansions, the PNC signal intensity in correspondence of the worker’s PBZ was much higher than the signal measured at 1.5 m from the source, in correspondence of the second worker. This confirms that the worst exposure condition was near the operator who directly performed the thermal expansion. In any case, the PNC reached noteworthy values, thus, it is reasonable to assume that a certain amount of the NPs that were released from the muffle furnace belonged to the produced material.

It should be noted that at the end of the three expansions carried out at lower temperature (phase 2A) the PNC was 2.5 times higher than the maximum value reached at the end of the phase 1A, even if during this phase the single peaks were higher. This can be related to the time consumption to carry out the different expansions in the two cases: the thermal expansion at a higher temperature was conducted within 12 min, whereas phase 2A was more time-consuming (22 min), increasing the final NPs concentration inside the laboratory.

The NPs release was confirmed by the SEM and EDS analysis conducted on the personal impactor sampled filters, which highligted the presence of some nanostructures with the same morphological features of the original material. The produced GNPs may be easily identified due to their particular shape (see Figure 1a), characterized by irregular and sharp edges and wedges. The morphological investigations revealed some airborne nanostructures, characterized by an irregular geometry, with the same shape and dimensions of the produced material. In the specific case, the nanostructures that were found on the Sioutas filters were materials cut-off from the WEGs main structure (see Figure 1b) during the thermal expansion. This could be noted from the aggregation level of WEGs pieces and their higher thickness compared to the GNPs.

As highlighted in the LDSA time series (Figure 5c and Figure 6c), this parameter followed the same trend of the PNC curves. In particular, during the thermal expansion at 1150 °C the LDSA reached higher values than during the thermal expansion at a lower temperature.

The NPs emitted during the thermal expansion were characterized by an average diameter ranging from 14 to 30 nm (phase 1A), with higher values recorded during the phase 2A (16.5 ÷ 37 nm), but in both cases they were lower than the NPs average diameter that characterized the background. These values were much lower than the lateral dimensions of GICs or WEGs and they can be attributed to the leakage from the muffle furnace of other non-carbonaceous nanostructures such as the expansion agents released from GICs during the thermal expansion, or NPs constituting the inner walls of the furnace (alumina) or even WEGs fragments detached from the main structure, that could have dimensions of a few nanometers. As a consequence, we suppose that the discrepancies between the modal average diameter measured by the real-time instruments and the size of WEGs or their fragments have been deeply influenced by these leakages from the muffle furnace.

In order to estimate the time needed to return the initial conditions at the end of each thermal expansion phase and to evaluate the possible differences between the two phases, we calculated the time decay of the PNC curve after opening the window (Table 4): this decay represents the time during which the NPs concentration goes below the significant value.

We observed that the time needed to return in the initial conditions after phase 1A was longer than the one after phase 2A. This can be be due to the different thermal expansion processes as well as to the different outdoor conditions that occurred at the end of each phase and could influence the indoor environment.

During both the liquid exfoliation phases there was no evidence of NPs release in lab 2 because the PNC time series show that, even if the values were higher than the corresponding bkg_FF_ measured in the adjacent lab 3, the general trend of the curve and the PNC mean values remained constant during the whole process, and they were lower than the significant value of the bkg_NF_. Moreover, the results of EDS and SEM analysis conducted on the Sioutas filters did not highlight the presence of any material attributable to those produced in this phase.

The LDSA values were a little higher than the bkg_FF_, due to the fact that this parameter was strictly related to the PNC (higher during the bkg_NF_ measurements), but in any case they could be considered as low values. The higher values of the bkg_FF_ measurements conducted before and during the liquid exfoliation were associated with events unrelated to the production.

During both the liquid exfoliation phases the average diameter was characterized by a mean value of 45 nm (phase 1B) and 55 nm (phase 2B) that were comparable with the background mean values measured in the same laboratory before starting the tip sonication. 

We could affirm that during each liquid exfoliation phase there was no workers’ exposure and it was probably due to the fact that the tip sonication was conducted by using a liquid solution under a chemical hood and the process was enclosed inside an ultrasonication shielding box. This is also confirmed by the results of the SEM analysis conducted on the sampled filters of the Sioutas placed just outside the ventilated chemical hood.

Although airborne NPs emission during the thermal expansion of GICs was identified, our study did not find relevant exposure to GNPs in the described production process. These results are in line with the findings reported in the majority of cases in literature [16,17]. The added value of our proposed analysis may be the comparison of exposure conditions in the thermal production of GNPs by exfoliation performed at different temperatures by the integration of real-time measurements and personal samplings for elemental and morphological analysis (SEM-EDS). Such a technique of microscopy has the advantage to investigate the morphology of airborne structures not modified during sample preparation, allowing us to characterize the real exposure scenario. Complementary studies by further characterization methods (e.g., elemental carbon analysis and TEM) [15] will be needed to identify the carbon-based materials emissions and the typical honeycomb lattice of graphene present in the personal samples.

## 5. Conclusions

In the present work we applied an integrated approach for workers’ exposure assessment during the production process of graphene nanoplatelets (GNPs) in a research laboratory. This methodology allowed us to compare different types of manufacturing processes including their single work phases, and to identify the process steps at a higher potential risk for workers, with the scope of proposing recommendation for a risk mitigation strategy. We performed real-time measurements before and during the single production phases by integrating different instruments together with off-line analysis (SEM and EDS analysis) on the filters sampled by personal impactors.

In the specific case, we studied two different exposure scenarios (Process 1 and Process 2) during which GNPs were produced, both consisting of two work phases: a thermal expansion in a muffle furnace of GICs (Phase A) and a liquid exfoliation of WEGs in acetone (Phase B). The two production processes differ in the thermal expansion phase: the first expansion was conducted at 1150 °C (phase 1A) and the second one at 1050 °C (phase 2A). The liquid exfoliation phase (phase 1B and phase 2B) was conducted in the same way during both the experimental campaigns.

At first, we investigated the background, by measuring the number concentration, the average diameter and the lung deposited surface area of the nanoparticles inside each laboratory before starting the expansion process (near field background). The far field background was measured inside an adjacent laboratory (lab 3) with volume, orientation and ventilation properties similar to the laboratories under study. During the second experimental campaign we monitored also the NPs concentration in a pre-chamber, next to the thermal expansion lab, in order to evaluate the nanoparticles values beside the production lab.

During both the experimental campaigns we carried out the measurements throughout the whole production process, comparing the results of the two phases and the results of the thermal expansion at 1150 °C with that performed at a lower temperature. Moreover, we compared the PNC values related to the production process with the bkg_FF_ measurements and the bkg_NF_ significant values, above which the concentration can be attributed to the production. We finally made a comparison between the personal values measured near the worker’s PBZ and the values obtained in correspondence of the position of a second worker, in order to verify the different exposure conditions. To estimate the time needed to return in safety conditions we also studied the time decay of the PNC after the end of the thermal expansion, during which we identified high exposure levels.

SEM and EDS analysis were performed on the sampled filters of the personal impactors, placed on the lab coat of the worker involved in the production activities, with the aim to confirm the possible nanoparticles release during the GNPs production.

The results of the real-time measurements and off-line analysis led to the following findings:During both the thermal expansion phases (phase 1A and phase 2A) we identified NPs release and possible workers’ exposure. After each expansion procedure the NPs concentration increased during the whole expansion process, reaching distinct peaks in correspondence to the opening of the furnace, which was greater than the selected bkg_NF_ significant values. During the whole expansion phase the NPs concentration was greater than the corresponding background far-field measurements.Looking at the PNC measurements performed during the thermal expansion by different real-time instruments located at different distances from the production source we could affirm that the personal exposure was higher than the workplace exposure. Furthermore, the measurements performed inside lab 1 and in the pre-chamber (beside the laboratory) confirm that in all the workstations the NPs concentration reached high values.The PNC time decay, representing the time needed to return in safety conditions after the thermal expansion, was different for the two phases but it was observed that was no less than ninety minutes.SEM and EDS analysis performed on the Sioutas filters during both the thermal expansion phases confirm that there was a release of carbonaceous nanostructures with irregular geometry and characterized by sharp edges and wedges, with the same shape of the fragments belonging to the material produced during this phase, i.e., WEGs. So, these nanostructures could be parts of WEGs who have detached during the thermal expansion from the main structure.Throughout both the liquid exfoliation phases (phase 1B and phase 2B) there was no evidence of any NPs released since the PNC values measured during the tip sonication were lower than the corresponding significant values. However, in front of the chemical hood the NPs concentration was higher than in correspondence of the sonicator control unit, confirming its efficacy as collective protective equipment.By comparison between two thermal expansions at a different temperature we could conclude that PNC peak levels in the phase 1A (1150 °C) were about 2.5 times greater than in phase 2A (1050 °C). Furthermore, the time needed to return in safety conditions after phase 1A was longer (about 1 h more) than phase 2A.

We could conclude that the worst exposure conditions arise during the thermal expansion of GICs near the operator and in the process performed at the highest temperature. 

The obtained structures (WEGs), with a nanoscale thickness, have larger lateral dimensions but they have the potential to cause adverse health effects in exposure by inhalation, due to their low aerodynamic diameter [14]. In any case, the use of personal protective equipment (PPE), as the full face respirators, the gloves and the lab coats, is justified due to the recognized exposure risk that characterize this phase. However, we may recommend to design this phase in a closed system and to organize the work taking into account the decay time of risk conditions inside the room after the end of the process.

Although there was no evidence of NPs released during the liquid exfoliation phase, it is nevertheless recommended to use PPE when handling the dry material while preparing the liquid suspension for the tip sonication, as it already happens.

In conclusion, the outcomes of this work provide important information in order to know in advance the risks for workers who produce graphene nanoplatelets by thermal expansion followed by a liquid exfoliation and to improve the future design of GNPs production process. The way forward will be to plan and select proper risk reduction measures by adopting a prevention-through-design approach.

## Figures and Tables

**Figure 1 nanomaterials-10-01520-f001:**
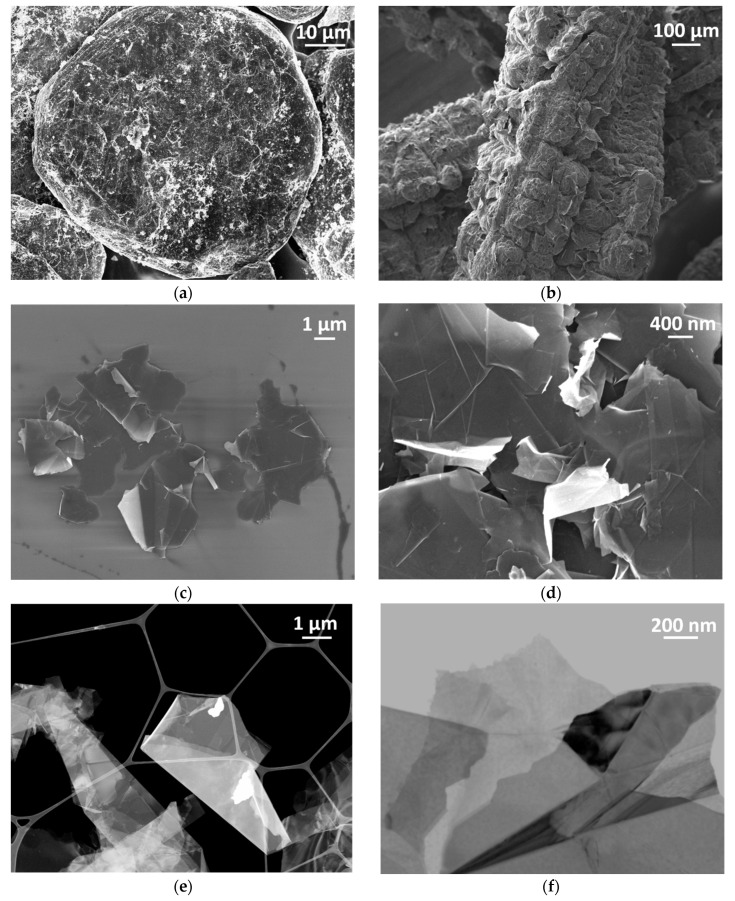
SEM images of the graphite intercalated compound (**a**), the worm-like expanded graphite (**b**) and graphene nanoplatelets’ (GNPs’) morphology (**c**,**d**) and STEM images of GNPs morphology (**e**,**f**).

**Figure 2 nanomaterials-10-01520-f002:**
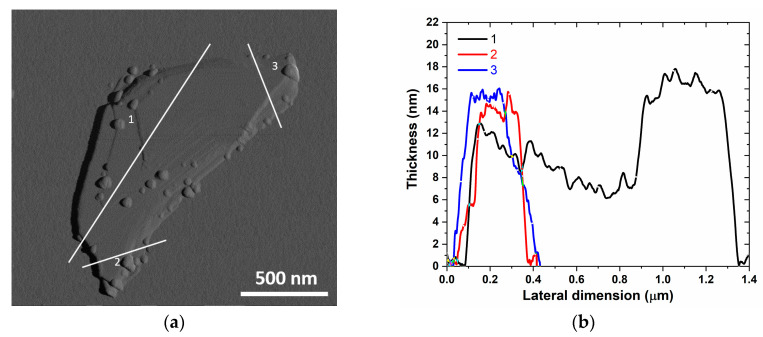
Atomic force microscopy (AFM) morphology of a GNP (**a**) and height profile along three different sections (**b**).

**Figure 3 nanomaterials-10-01520-f003:**
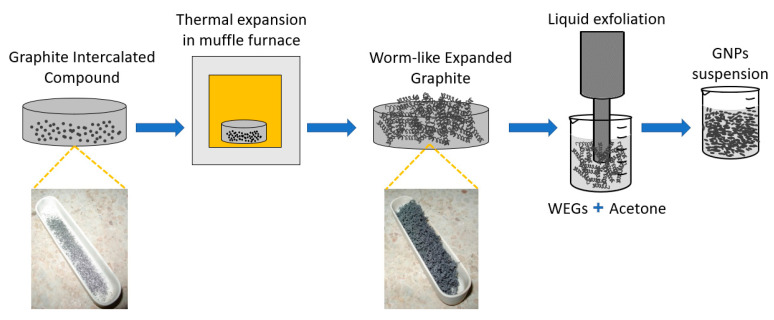
Scheme of the GNPs production process.

**Figure 4 nanomaterials-10-01520-f004:**
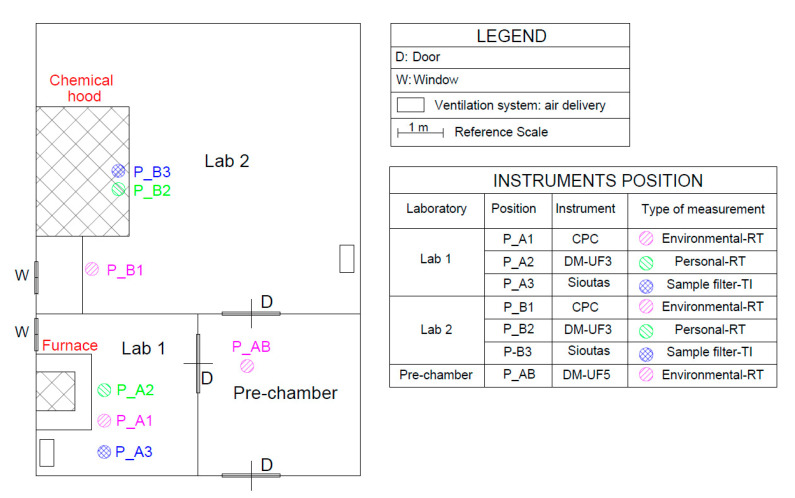
Schematic layout of the production labs.

**Figure 5 nanomaterials-10-01520-f005:**
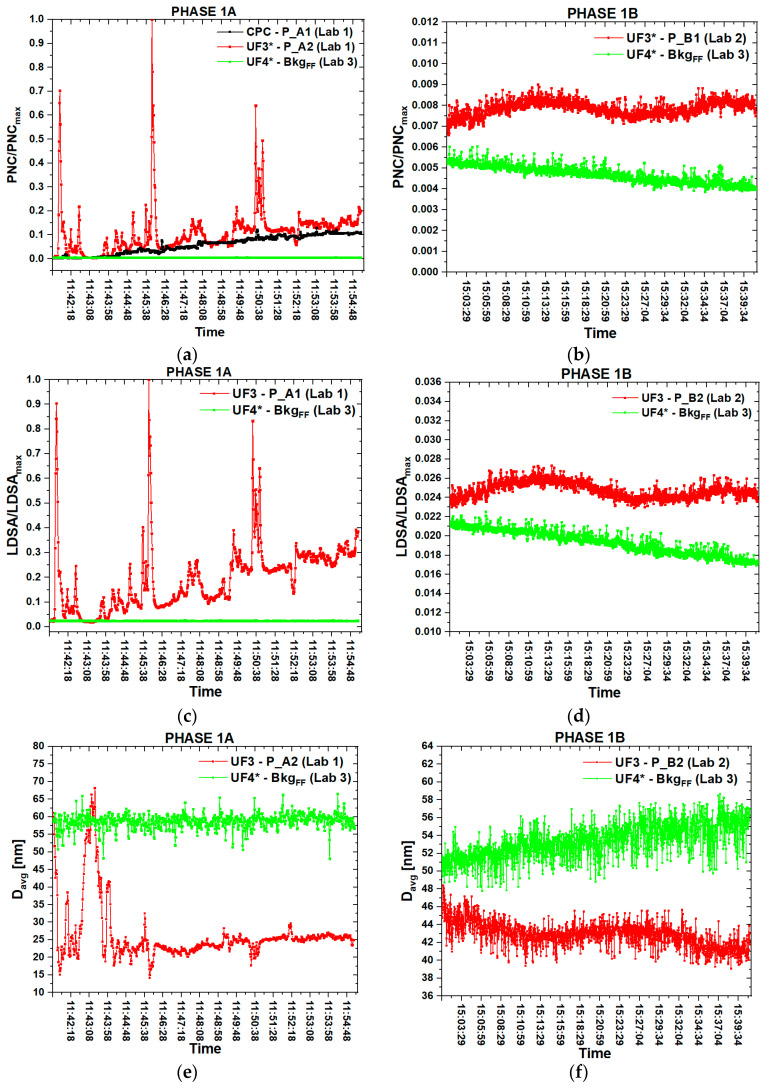
Time series of particle number concentration (PNC; **a**), lung deposited surface area (LDSA; **c**) and modal average diameter (D_avg_; **e**) referred to the thermal expansion at 1150 °C (phase 1A) and time series of PNC (**b**), LDSA (**d**) and D_avg_ (**f**) referred to the liquid exfoliation (phase 1B).

**Figure 6 nanomaterials-10-01520-f006:**
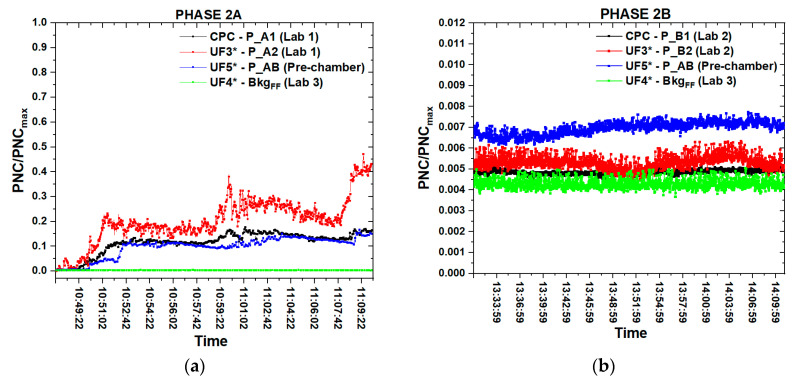
Time series of PNC (**a**), LDSA (**c**) and D_avg_ (**e**) referred to the thermal expansion at 1050 °C (phase 2A) and time series of PNC (**b**), LDSA (**d**) and D_avg_ (**f**) referred to the liquid exfoliation (phase 2B).

**Figure 7 nanomaterials-10-01520-f007:**
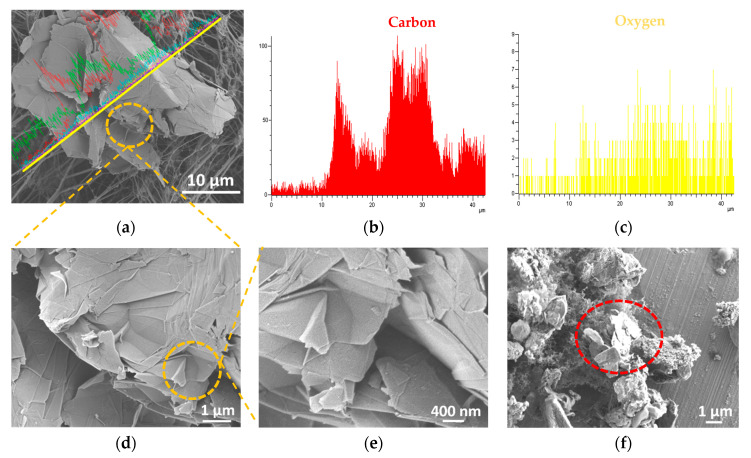
EDS spectrum of the material collected during the thermal expansion process by the Sioutas on the backup filter (**a**–**c**), two magnifications of a SEM image of the material collected on the backup filter (**d**,**e**) and a SEM image of the filter C (**f**) with “worm-like” expanded graphites (WEGs) highlighted in the red circle.

**Figure 8 nanomaterials-10-01520-f008:**
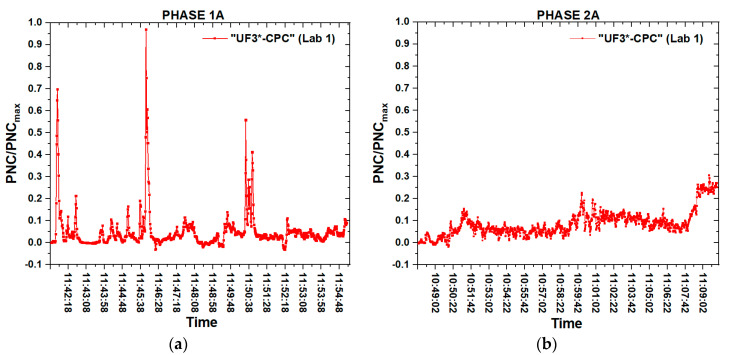
Residual curve of phase 1A (**a**) and phase 2A (**b**) as the difference between the PNC time series of the personal and the environmental instrument.

**Table 1 nanomaterials-10-01520-t001:** Timesheet of the measurements during the first (**a**) and the second (**b**) exposure’s campaigns.

**(a)**			
**DAY**	**HOUR**	**MEASUREMENTS**	**LABS**
DAY 1: Parallel and background measurements	01:00 p.m.–01:40 p.m.	Parallel and bkg_NF_	Lab 1
	01:45 p.m.–02:45 p.m.	Parallel and bkg_NF_	
	03:00 p.m.–03:35 p.m.	Parallel and bkg_NF_	Lab 2
	03:45 p.m.–04:15 p.m.	Parallel and bkg_FF_	Lab 3
DAY 2: Measurements during the GNPs production process	10:05 a.m.–11:20 a.m.	bkg_FF_ bkg_NF_	Lab 3 Lab 1
	11:30 a.m.–11:40 a.m.	bkg_FF_ bkg_NF_-pre	Lab 3 Lab 1
	11:40 a.m.–11:55 a.m.	Thermal expansion (Phase 1A)	Lab 1
	11:55 a.m.–02:25 p.m.	bkg_NF_	
	02:30 p.m.–02:55 p.m.	bkg_NF_	Lab 2
	02:55 p.m.–03:40 p.m.	Liquid exfoliation (Phase 1B)	
	03:40 p.m.–04:00 p.m.	bkg_NF_	
	04:15 p.m.–04:50 p.m.	Parallel and bkg_NF_-post	
**(b)**			
**DAY**	**HOURS**	**MEASUREMENTS**	**LABS**
DAY 1: Parallel and background measurements	11:45 a.m.–12:30 a.m.	Parallel and bkg_NF_	Lab 1
	12:40 a.m.–13:30 p.m.	Parallel and bkg_NF_	
	13:50 a.m.–02:40 p.m.	Parallel and bkg_NF_	Lab 2
	02:40 a.m.–03:50 p.m.	Parallel and bkg_FF_	Pre-chamber
	04:00 p.m.–09:30 a.m.	Night bkg_NF_	Lab 1 Pre-chamber Lab 3
DAY 2: Measurements during the GNPs production process	09:30 a.m.–10:30 a.m.	Parallel and bkg_NF_-pre	Lab 1 Pre-chamber
	10:40 a.m.–01:00 p.m.	Thermal expansion (Phase 2A)	Lab 1 Pre-chamber
	01:20 p.m.–02:20 p.m.	Liquid exfoliation (Phase 2B)	Lab 2 Pre-chamber
	02:30 p.m.–03:20 p.m.	Parallel and bkg_NF_-post	Lab 1

**Table 2 nanomaterials-10-01520-t002:** Characteristic values of the background of phase 1A (**a**) and 1B (**b**).

(**a**)							
	**Real-Time Instrument**	**Parameter**	**Mean**	**Σ**	**Minimum**	**Median**	**Maximum**
THERMAL EXPANSION (PHASE 1A)	CPC-bkg_NF_ (P_A1)	PNC (#/cm^3^)	7589	199	7002	7607	8165
	DM-UF3-bkg_NF_ (P_A2)	PNC (#/cm^3^)	8194	955	4624	8034	11,244
		D_avg_ (nm)	68.7	4.5	53.2	68.8	90.7
		LDSA (µm^2^/cm^3^)	28.6	2.3	24.3	27.8	37.9
	DM-UF4-bkg_FF_ (Lab 3)	PNC (#/cm^3^)	8912	669	7679	8799	10,889
		D_avg_ (nm)	59.0	2.1	51.1	59.1	66.5
		LDSA (µm^2^/cm^3^)	29.2	1.6	26.2	28.9	33.9
(**b**)							
	**Real-Time Instrument**	**Parameter**	**Mean**	**Σ**	**Minimum**	**Median**	**Maximum**
LIQUID EXFOLIATION (PHASE 1B)	CPC-bkg_NF_ (P_B1)	PNC (#/cm^3^)	12,213	410	11,376	12,135	13,254
	DM-UF3-bkg_NF_ (P_B3)	PNC (#/cm^3^)	12,303	492	11,361	12,214	13,841
		D_avg_ (nm)	45.8	1.0	42.0	45.9	49.1
		LDSA (µm^2^/cm^3^)	28.8	0.9	26.9	28.6	31.5
	DM-UF4-bkg_FF _ (Lab 3)	PNC (#/cm^3^)	9300	302	8467	9291	10,108
		D_avg_ (nm)	49.3	1.3	44.7	49.3	54.4
		LDSA (µm^2^/cm^3^)	25.1	0.6	23.6	25.0	26.9

**Table 3 nanomaterials-10-01520-t003:** Characteristic values of the background measurements of phase 2A (**a**) and 2B (**b**).

(**a**)							
	**Real-Time Instruments**	**Parameter**	**Mean**	**σ**	**Minimum**	**Median**	**Maximum**
THERMAL EXPANSION (PHASE 2A)	CPC-bkg_NF_ (Lab 1-P_A1)	PNC (#/cm^3^)	9273	318	8540	9214	10,630
	UF3-bkg_NF_ (Lab 1-P_A2)	PNC (#/cm^3^)	11,420	800	9595	11,332	14,312
		D_avg_ (nm)	51.6	1.9	46.4	51.5	58.0
		LDSA (µm^2^/cm^3^)	25.3	1.1	23.5	25.1	30.5
	UF5-bkg_NF_ (Pre-chamber-P_AB)	PNC (#/cm^3^)	10,000	176	9534	9990	10,627
		D_avg_ (nm)	46.7	1.2	43.5	47.7	50.2
		LDSA (µm^2^/cm^3^)	22.3	0.9	20.6	22.4	24.7
	UF4-bkg_FF_ (Lab 3)	PNC (#/cm^3^)	6842	350	5957	6795	8333
		D_avg_ (nm)	45.1	1.2	40.0	45.1	49.7
		LDSA (µm^2^/cm^3^)	15.9	0.6	14.4	15.9	17.6
(**b**)							
	**Real-Time Instruments**	**Parameter**	**Mean**	**σ**	**Minimum**	**Median**	**Maximum**
LIQUID EXFOLIATION (PHASE 2B)	CPC-bkg_NF_ (Lab2-P_B1)	PNC (#/cm^3^)	7710	188	7193	7699	8234
	UF3-bkg_NF_ (Lab2-P_B3)	PNC (#/cm^3^)	8264	586	6977	8164	10,897
		D_avg_ (nm)	55.2	2.2	46.7	55.6	61.6
		LDSA (µm^2^/cm^3^)	20.9	0.5	19.8	20.8	22.4
	UF5-bkg_NF_ (Pre-chamber-P_AB)	PNC (#/cm^3^)	10,326	317	9419	10,322	11,371
		D_avg_ (nm)	46.5	1.2	38.7	46.6	51.7
		LDSA (µm^2^/cm^3^)	23.2	0.7	21.7	23.2	25.5
	UF4-bkg_FF_ (Lab 3)	PNC (#/cm^3^)	6858	322	6152	6776	8365
		D_avg_ (nm)	46.3	1.5	40.7	46.8	52.3
		LDSA (µm^2^/cm^3^)	16.6	0.2	16.0	16.5	17.4

**Table 4 nanomaterials-10-01520-t004:** Time decay of the PNC curves after the thermal expansion phases.

Phase	Time (Max Value)	Time (Min Value)	Time Decay
Phase 1A	11:55:03 a.m.	02:24:11 p.m.	2 h 30 min
Phase 2A	11:10:03 a.m.	12:37:47 a.m.	1 h 30 min

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
