# Peer review of "Workers’ Exposure Assessment during the Production of Graphene Nanoplatelets in R&D Laboratory"

_nanomaterials, 2020, doi:10.3390/nano10081520_

Round 1
Reviewer 1 Report
This work report workers’ exposure assessment during the production of Graphene Nanoplatelets in R&D Laboratory. It is of interest to many, and I think it should be published in a journal that is widely read. One thing that I am not sure about is the level of representative condition of the study. The conclusion that the risk is higher during the thermal expansion of GICs
near the operator and in the process performed at the highest temperature. However, I think the above conclusion may vary if the initial condition of the R & D lab is different, can the authors comment on that?
Author Response
Authors thank the reviewer for his valuable observation and comment. In the following, the answer to reviewer’s comment is reported.
Point 1. One thing that I am not sure about is the level of representative condition of the study. The conclusion that the risk is higher during the thermal expansion of GICs near the operator and in the process performed at the highest temperature. However, I think the above conclusion may vary if the initial condition of the R & D lab is different, can the authors comment on that?
Response 1: In our study we repeated two times the same production of GNPs by thermal exfoliation to compare the exposure when the process was conducted at different temperature. In two cases we started from slightly different background conditions (both FF and NF) as reported in Table 2 and 3 in the manuscript, but we noted a similar general trend of PNC, in particular during the thermal expansion with three clear peaks in correspondence of each expansion in both cases and simultaneous average size decrease. Based on the differences related to the personal exposure conditions as reported in Figure 7 we concluded that “the risk is higher during the thermal expansion of GICs near the operator and in the process performed at the highest temperature”.
We agree with the reviewer comment and we know that the background conditions of the laboratory may strongly influence the exposure values in any case. To improve the statistical significance of the result we need to repeat more time the experiment in the same boundary conditions and make it representative. Often this may be not cost-effective and in contrast with the production needs of the production laboratory.

Reviewer 2 Report
The author reports the application of a multi-metric approach to access worker’s exposure during the production of graphene nanoplatelets by thermal exfoliation.
Major issues:
May be the author can consider to shorten the abstract to make it more precise and around 200 words, instead of about 274 words for the current manuscript.
Please, if possible, to include the safety levels/standards that is recommended by government bodies, or other international safety associations (e.g., ASTM international standard). For example, the particle number concentration readings acquired by the condensation particle counter and mini diffusion size classifier, which are considered to be safe, should be given to allow the general reader for comparison. Other measurement given in this manuscript should also be compared with the safety standards.
In materials and methods section, the author may consider revising the scanning electron microscope images of graphite intercalated compounds in the result section. Moreover, more description of the laboratory and personel should be given. For example, the laboratory is air-conditioned, or without air conditioned, windows/doors is/are opened/closed should be given, and operation were carried out by same two workers. These variations may affect the results, thus, may need more than one tests (in the result section) to investigate on the changes due to these factors, or other unforeseen and unknown factors.
In the discussion section, results from other nanoparticles safety level investigations in the literature should be compared and contrast.
In conclusion section, the author may think of revising some parts to be included in the results, or discussion section, in order to make the conclusion to be more summarized, precise and conclusive.
Minor issues:
Line 28, should be “approach to ‘assess’ workers’ exposure…”
Line 91, please check if the author wants to describe “worm-like” instead of “warm-like”.
Please check again that several reference bookmarks cannot be found. For example, line 190, “errore. l'autoriferimento non è valido per un segnalibro….”
Please be careful with abbreviations. The exact word/phrase should be given and explain about the abbreviations at least once for the first occurrence. For example, SEM, STEM, FE-SEM, environment-RT, sample filter-TI. These abbreviations may be obvious to the author, but can be difficult to understand for the general readers.
Author Response
Authors thank the reviewer for his valuable observations and comments. In the following, the answers to reviewer’s comments is reported.
Major issues:
Point 1: May be the author can consider to shorten the abstract to make it more precise and around 200 words, instead of about 274 words for the current manuscript.
Response 1: We have reduced the abstract to 207 words.
“Widespread production and use of engineered nanomaterials in industrial and research settings raise concerns about their health impact in the workplace. In last years, graphene-based nanomaterials have gained particular interest in many application fields. Among them, graphene nanoplatelets (GNPs) showed superior electrical, optical and thermal properties, low-cost and availability. Few and conflicting results have been reported about toxicity and potential effects on workers’ health, during the production and handling of these nanostructures. Due to this lack of knowledge, systematic approaches are needed to assess risks and quantify workers’ exposure to GNPs. This work applies a multi-metric approach to asses workers’ exposure during the production of GNPs, based on the OECD methodology by integrating real-time measurements and personal sampling. In particular, we analysed the particle number concentration, the average diameter and the Lung Deposited Surface Area of airborne nanoparticles during the production process conducted by thermal exfoliation in two different ways, compared to the background. These results have been integrated by electron microscopic and spectroscopic analysis on the filters sampled by personal impactors. The study identifies the process phases potentially at risk for workers and reports quantitative information about the parameters that may influence the exposure in order to propose recommendations for a safer design of GNPs production process.”
Point 2: Please, if possible, to include the safety levels/standards that is recommended by government bodies, or other international safety associations (e.g., ASTM international standard). For example, the particle number concentration readings acquired by the condensation particle counter and mini diffusion size classifier, which are considered to be safe, should be given to allow the general reader for comparison. Other measurement given in this manuscript should also be compared with the safety standards.
Response 2: At present, occupational exposure limits or safety standard levels recommended by government bodies for graphene nanoplatelets exposure are not available. We found in literature some reviews and studies that report measured values of particle number concentration and average size during the GNP production.
Spinazzè et al. [17] assessed the effectiveness of fume hoods in protecting the operators against graphene exposure and to define the conditions that result in lower dispersion of particles in the chemical hood and in the laboratory environment. They conducted simulations by the handling of commercial GNPs with dimensions of a few micrometers and thickness of a few nanometers inside a hood and they measured real-time PNC and LDSA, but not integrated by analysis of composition of airborne materials.
Different production methods of graphite exfoliation and chemical vapour deposition were tested by Lee et al. [15] using elemental carbon analysis, real-time measurements and TEM analysis which allows the identification of the typical structure of honeycomb graphene lattice but requires necessarily the modification of sampled airborne structures.
In general, as reported in the review by the Fadeel et al [16], the available studies reported not relevant or low exposure to graphene-based materials in the production conditions.
Spinazzè et al. 2018 in their “Probabilistic approach for the risk assessment of nanomaterials: A case study for graphene nanoplatelets” performed a literature search regarding occupational exposure assessments for GNPs, and they selected three studies for their analysis, including the previously cited Spinazzè et al. [17] and Lee at al. [15]. They concluded that workplaces studied in the three studies were strongly influenced by the presence of other particles than GNPs.
Finally, we do not find any paper in which GNPs were produced by thermal exfoliation, then a fully comparison of our findings with other studies is challenging, since it has been recognized that the production method and the workplace conditions may strongly influence the exposure.
Based on such consideration we added the following paragraph in the Discussion section Page 16 (lines 468-478) for general reader to allow comparison:
“Although airborne NPs emission during the thermal expansion of GICs was identified, our study does not find relevant exposure to GNP in the described production process. These results are in line with the findings reported in the majority of cases in literature [16], [17]. The added value of our proposed analysis may be the comparison of exposure conditions in the thermal production of GNPs by exfoliation performed at different temperature by the integration of real-time measurements and personal samplings for elemental and morphological analysis (SEM-EDS). Such technique of microscopy has the advantage to investigate the morphology of airborne structures not modified during sample preparation, allowing us to characterize the real exposure scenario. Complementary studies by further characterization methods (e.g. elemental carbon analysis and TEM) [15] will be needed to identify the carbon-based materials emissions and the typical honeycomb lattice of graphene present in the personal samples.”
Point 3.1: In materials and methods section, the author may consider revising the scanning electron microscope images of graphite intercalated compounds in the result section.
Response 3.1: We are sorry, but we do not understand this comment. In any case we improved the materials characterization also to reply to another reviewer, including also the AFM analysis. In the discussion section we already cited the Fig. 1b for comparison to the airborne sampled materials. Due to the very short deadline to send revised paper we are not able to conduct more SEM analysis on sampled filters. In any case we will be available to further improve the materials and methods section or the results section if you could clarify the request.
Point 3.2: Moreover, more description of the laboratory and personnel should be given. For example, the laboratory is air-conditioned, or without air conditioned, windows/doors is/are opened/closed should be given, and operation were carried out by same two workers. These variations may affect the results, thus, may need more than one tests (in the result section) to investigate on the changes due to these factors, or other unforeseen and unknown factors.
Response 3.2: We agree with the reviewer comment and we know that the ventilation conditions of the laboratory may strongly influence the exposure values in any case. As from Fig. 4 the laboratories are air conditioned and windows/doors were closed during the activities. Only in the Lab 1, at the end of the thermal expansion phase the window has opened. We know that to improve the statistical significance of the result we need to repeat more time the experiment in the same boundary conditions and make it representative. Often this may be not cost-effective and in contrast with the production needs of the production laboratory.
We added the following sentences:
1) (Page 6 lines 202-203): “As from Figure 4 the laboratories are air conditioned and windows/doors were closed during the activities. Only in the lab 1, at the end of the thermal expansion phase the window has opened.”
2) (Page 5 lines 137-140): “The production activities were carried out by the same two workers in both the experimental campaigns.”
In our study we repeated two times the same production of GNPs by thermal exfoliation to compare the exposure when the process was conducted at different temperature. In two cases we started from slightly different background conditions (both FF and NF) as reported in Table 2 and 3 in the manuscript, and we noted a similar general trend of PNC, in particular during the thermal expansion with three clear peaks in correspondence of each expansion in both cases and simultaneous average size decrease.
Point 4: In the discussion section, results from other nanoparticles safety level investigations in the literature should be compared and contrast.
Response 4: We agree with the reviewer comment and we know that the ventilation conditions of the laboratory may strongly influence the exposure values in any case. As from Fig. 4 the laboratories are air conditioned and windows/doors were closed during the activities. Only in the Lab 1, at the end of the thermal expansion phase the window has opened. We know that to improve the statistical significance of the result we need to repeat more time the experiment in the same boundary conditions and make it representative. Often this may be not cost-effective and in contrast with the production needs of the production laboratory.
In our study we repeated two times the same production of GNPs by thermal exfoliation to compare the exposure when the process was conducted at different temperature. In two cases we started from slightly different background conditions (both FF and NF) as reported in Table 2 and 3 in the manuscript, and we noted a similar general trend of PNC, in particular during the thermal expansion with three clear peaks in correspondence of each expansion in both cases and simultaneous average size decrease.
Point 5: In conclusion section, the author may think of revising some parts to be included in the results, or discussion section, in order to make the conclusion to be more summarized, precise and conclusive.
Response 5: According to the authors guidelines the Conclusions section “is not mandatory but can be added to the manuscript if the discussion is unusually long or complex.” We decided to include also the conclusion section in order to summarize the major results (in bullet points) and to give some safety recommendations. We feel that this may help the reader to highlight the major finding of our complex study and we would like to keep it in this form. In any case, if also the Editor in chief do not agree with this choice, we will be able to eliminate this section and reformulate the discussion.
Point 6: Line 28, should be “approach to ‘assess’ workers’ exposure…”
Response 6: We have corrected the typos.
Point 7: Line 91, please check if the author wants to describe “worm-like” instead of “warm-like”.
Response 7: We corrected the typos.
Point 8: Please check again that several reference bookmarks cannot be found. For example, line 190, “errore. l'autoriferimento non è valido per un segnalibro….”
Response 8: We are sorry for the compilation error; We have adjusted the references linked to the tables and the figures.
Point 9: Please be careful with abbreviations. The exact word/phrase should be given and explain about the abbreviations at least once for the first occurrence. For example, SEM, STEM, FE-SEM, environment-RT, sample filter-TI. These abbreviations may be obvious to the author but can be difficult to understand for the general readers.
Response 9: We have introduced the abbreviations in the first occurrence.

Reviewer 3 Report
Please see attached file.

Author Response
Authors thank the reviewer for his valuable and constructive comments and suggestions. We have carefully analyzed them and consequently we have improved some parts of the manuscript. Changes are marked in yellow in the revised manuscript. In the following, the list of changes and answers to the reviewer’s comments are reported.
Point 1. Page 3, lines 94-96: “As shown in the SEM images reported in Figure 1 (c, d) GNPs have lateral dimensions in the micrometer-range (between 1 micron and 5 micron) and an average thickness varying from 1 nm up to 10 nm, they are characterized by an irregular geometry and by sharp edges and wedges.” SEM is not a suitable technique for measuring GNP thickness, other characterization techniques (in particular, AFM) are more appropriated for this purpose. Please comment on this issue. It also would be nice if the authors provide, if possible, information on the textural properties (surface area, pore size and type, …) of the produced GNPs.
Response 1: Thank you very much for your suggestion. We performed an AFM analysis in order to show the average thickness of the graphene nanoplatelets and the average surface area. Furthermore, we added another reference referred to our previous work in which we assessed other detailed studies on the GNPs morphology and size distribution [24]. We added the following sentence (pages 3-4, lines 101-105):
“In order to evaluate the thickness of the produced GNPs, we assessed an Atomic Force Microscopy (AFM) analysis, using a Veeco ICON available in the Nanotechnology and Nanoscience Laboratory of Sapienza University (NNS-Lab). Figure 2 (a) shows an AFM image of a single GNP with stacked graphene sheets. The measured profile of this structure (Figure 2 (b)) shows a thickness varying from ~7 nm to ~18 nm [24]. The GNP lateral dimensions are of around 1 µm2 and the average surface area is of ~1.5 μm.”
Furthermore, we modified the sentence in page 3 lines 96-98 as follows:
“As shown in the SEM images reported in Figure 1 (c, d) GNPs have lateral dimensions in the micrometre-range (between 1 μm and 5 μm) and they are characterized by an irregular geometry and by sharp edges and wedges.”
[24] I. Rago et al., “Antimicrobial activity of graphene nanoplatelets against Streptococcus mutans,” in 2015 IEEE 15th International Conference on Nanotechnology (IEEE-NANO), 2015, pp. 9–12.
Point 2. There is a discrepancy between the measured modal average diameter (Davg) values and the size of the produced GNPs. This issue should be explained in the manuscript for clarity purposes. This is important also because really small carbon clusters are ejected during the production of certain carbon nanomaterials, but are detected using other characterization techniques, different to those reported here.
Response 2: In the paper we highlighted this issue by the sentence in page 15 lines 427-434, that has been modified as follows in order to clarify the concept:
“The NPs emitted during the thermal expansion are characterized by an average diameter ranging from 14 nm to 30 nm (phase 1A), (…). These values are much lower than the lateral dimensions of GICs or WEGs and they can be attributed to the leakage from the muffle furnace of other non-carbonaceous nanostructures such as the expansion agents released from GICs during the thermal expansion, or NPs constituting the inner walls of the furnace (alumina) or even WEGs fragments detached from the main structure, that could have dimensions of few nanometers.”
Furthermore, we added the following sentence in order to underline the abovementioned discrepancy (Page 15-16 lines 434-436):
“As consequence, we suppose that the discrepancies between the modal average diameter measured by the real-time instruments and the size of WEGs or their fragments have been deeply influenced by these leakages from the muffle furnace.”
Point 3. Please indicate the units of the parameters presented in Table 2.
Response 3: We are sorry for this negligence; we added the units of those parameters in table 2 and in table 3.
Point 4. Page 13, lines 365 to 370, and Page 16, lines 498 to 502: “SEM and EDS analyses confirm that during both the thermal expansion phases there is a release of carbonaceous nanostructures with irregular geometry and characterized by sharp edges and wedges, with the same shape and dimensions of the produced material. These nanostructures (sic) could be parts of WEGs who (sic) have detached during the thermal expansion from the main structure”. This is a bit confusing: are the materials collected at the Sioutas only WEG fragments ejected during thermal expansion procedures? According to the personal impactor’s characteristics (see page 5), it should be able to collect smaller airborne particles, such as GNPs. Please comment on the carbon materials collected at the Sioutas.
Response 4: Sorry for the confused sentences. I’d like to highlight that this comment is referred to the SEM analysis performed on the filters collected during the thermal expansion of GICs. So we expected to see only WEGs fragments, that have been detached from the main bigger structure, characterized by the same GNPs morphology (sharp edges and wedges) but with higher lateral dimensions and thickness.
We reworded the two sentences in order to make them more clear:
1) Page 14 lines 380-386: “The morphological investigations reported in Figure 6 (d-f) reveal some nanostructures that are characterized by an irregular geometry and by sharp edges and wedges, with the same shape of the fragments belonging to the material produced during the thermal expansion, i.e. WEGs (see Figure 1(b)). In particular, the EDS analysis (Figure 6 (a-c)) reveal the carbon signal, that can feel like the chemical composition of WEGs, made up of only carbon atoms.
From the SEM analysis conducted on the filters collected during the liquid exfoliation, there is not evidence of any material attributable to this phase, i.e. GNPs.”
2) Page 17 lines 525-530: ”SEM and EDS analysis performed on the Sioutas filters during both the thermal expansion phases confirm that there is a release of carbonaceous nanostructures with irregular geometry and characterized by sharp edges and wedges, with the same shape of the fragments belonging to the material produced during this phase, i.e. WEGs. So, these nanostructures could be parts of WEGs who have detached during the thermal expansion from the main structure.”
Point 5. Please compare/correlate the results shown here to those already reported in the literature (see for example references 14 to 16, as well as A. Spinazzè et al., “Probabilistic approach for the risk assessment of nanomaterials: A case study for graphene nanoplatelets”, International Journal of Hygiene and Environmental Health, vol. 222, pp. 76- 83, 2019) and highlight the contribution of this work in the frame of those already published studies. This is an important issue as the structural features of the GNPs may vary depending on the production process used (in particular, the size of the GNPs described in the present manuscript are different than other commercial GNP materials, that sometimes consist of large aggregates of significantly smaller platelets).
Response 5: Spinazzè et al. [17] assessed the effectiveness of fume hoods in protecting the operators against graphene exposure and to define the conditions that result in lower dispersion of particles in the chemical hood and in the laboratory environment. They conducted simulations by the handling of commercial GNPs with dimensions of a few micrometers and thickness of a few nanometers inside a hood and they measured real-time PNC and LDSA, but not integrated by analysis of composition of airborne materials.
Different production methods of graphite exfoliation and chemical vapour deposition were tested by Lee et al. [15] using elemental carbon analysis, real-time measurements and TEM analysis which allows the identification of the typical structure of honeycomb graphene lattice but requires necessarily the modification of sampled airborne structures.
In general, as reported in the review by the Fadeel et al [16], the available studies reported not relevant or low exposure to graphene-based materials in the production conditions.
Spinazzè et al. 2018 in their “Probabilistic approach for the risk assessment of nanomaterials: A case study for graphene nanoplatelets” performed a literature search regarding occupational exposure assessments for GNPs, and they selected three studies for their analysis, including the previously cited Spinazzè et al. [17] and Lee at al. [15]. They concluded that workplaces studied in the three studies were strongly influenced by the presence of other particles than GNPs.
Finally, we do not find any paper in which GNPs were produced by thermal exfoliation, then a fully comparison of our findings with other studies is challenging, since it has been recognized that the production method and the workplace conditions may strongly influence the exposure.
Based on such consideration we added the following paragraph in the Discussion section Page 16 (lines 465-475):
“Although airborne NPs emission during the thermal expansion of GICs was identified, our study does not find relevant exposure to GNPs in the described production process. These results are in line with the findings reported in the majority of cases in literature [16], [17]. The added value of our proposed analysis may be the comparison of exposure conditions in the thermal production of GNPs by exfoliation performed at different temperatures by the integration of real-time measurements and personal samplings for elemental and morphological analysis (SEM-EDS). Such technique of microscopy has the advantage to investigate the morphology of airborne structures not modified during sample preparation, allowing us to characterize the real exposure scenario. Complementary studies by further characterization methods (e.g. elemental carbon analysis and TEM) [15] will be needed to identify the carbon-based materials emissions and the typical honeycomb lattice of graphene present in the personal samples.”
[15] J. H. Lee et al., “Exposure monitoring of graphene nanoplatelets manufacturing workplaces,” Inhal. Toxicol., vol. 28, no. 6, pp. 281–291, 2016.
[16] B. Fadeel et al., “Safety Assessment of Graphene-Based Materials: Focus on Human Health and the Environment,” ACS Nano, vol. 12, no. 11, pp. 10582–10620, 2018.
[17] A. Spinazzè et al., “Exposure to airborne particles associated with the handling of graphene nanoplatelets,” Med Lav., vol. 109, no. 4, pp. 285–296, 2018.
Point 6. In the Discussion and the Conclusions sections I miss a statement of the hazard level of the GNP production and handling processes described here in light of the different parameters measured in this work. This is expected of a paper that addresses working conditions that may affect the researcher’s health.
Response 6: We reported in the Introduction some general findings extracted by available studies on the exposure to graphene based NMs, that report conflicting outcomes about their hazardous properties [11] – [14].
In particular K.H. Liao et al. [11] demonstrate that “particle size, particulate state, and oxygen content/surface charge of graphene have a strong impact on biological/toxicological responses to red blood cells”, concluding that the toxicity of graphene “may depend on the exposure environment (i.e., whether or not aggregation occurs) and mode of interaction with cells (i.e., suspension versus adherent cell types)”.
The literature review by Ou et al. [12] highlighted that “…many experiments have shown that GFNs have toxic side effects in many biological applications, but the in-depth study of toxicity mechanisms is urgently needed. In addition, contrasting results regarding the toxicity of GFNs need to be addressed by effective experimental methods and systematic studies”.
The review by C. Liao et al. [13] highlighted that the production method plays a key role in the toxicity evaluation of graphene based NMs: different production methods may produce different impacts on biocompatibility and cytotoxicity.
Finally Schinwald et al. 2012 [14] stressed the importance of the size and shape of 2-d NMs in all toxicity evaluations, with particular reference to their aerodynamic diameter for exposure by inhalation: They suggested that “sheet/platelet-shaped particles with nanoscale thickness can be very large in two dimensions but possess a low aerodynamic diameter. Thus, they have potential to cause adverse health effects”.
Since our study is not focused on toxicological effects of GNPs, we have decided to not report this detailed analysis in the paper. In any case, based on your suggestions, we modified the Conclusions section including the following sentence (page 18 lines 542-543):
“Such structures with nanoscale thickness have larger lateral dimensions but they have potential to cause adverse health effects in exposure by inhalation, due to their low aerodynamic diameter [14].”
[11] K.-H. Liao, Y.-S. Lin, C. W. Macosko, and C. L. Haynes, “Cytotoxicity of Graphene Oxide and Graphene in Human Erythrocytes and Skin Fibroblasts,” ACS Appl. Mater. Interfaces, vol. 3, no. 7, pp. 2607–2615, Jul. 2011.
[12] L. Ou et al., “Toxicity of graphene-family nanoparticles: a general review of the origins and mechanisms,” Part. Fibre Toxicol., vol. 13, no. 1, p. 57, Dec. 2016.
[13] C. Liao, Y. Li, and S. Tjong, “Graphene Nanomaterials: Synthesis, Biocompatibility, and Cytotoxicity,” Int. J. Mol. Sci., vol. 19, no. 11, p. 3564, Nov. 2018.
[14] A. Schinwald, K. Donaldson, F. A. Murphy, A. Jones, and W. MacNee, “Graphene-Based Nanoplatelets: A New Risk to the Respiratory System as a Consequence of Their Unusual Aerodynamic Properties,” ACS Nano, vol. 6, no. 1, pp. 736–746, 2012.
Point 7. Acetone was used as solvent for the probe-sonication-assisted liquid exfoliation process. Probe-sonication is known to induce acetone evaporation, that eventually may be a hazard itself and a safety issue. Have the authors addressed this issue as well? Also, would it be beneficial for the workers’ safety if GNP production, handling and liquid exfoliation experiments were performed in well ventilated lab environments (for example, operating with opened windows, see Figure 3), as it is usually recommended?
Response 7: We agree with the reviewer comment, in fact in order to avoid the evaporation of the solvent, despite the liquid exfoliation takes place under a ventilated chemical hood, we performed this phase by using a cold jacketed beaker, maintaining the GNPs/acetone solution at a constant temperature of 5°C, minimizing the risk of workers’ exposure during this phase. For this reason, we didn’t address this issue.
We modified the sentence in page 5, lines 138-139, in order to specify the liquid exfoliation phase operative condition, as follows (page 5 lines 133-136):
1) “The tip sonication is performed under a chemical ventilated hood at room temperature for 40 minutes by using a cold jacketed beaker, maintaining the GNPs/acetone solution at a constant temperature of 5°C, thus avoiding the possible evaporation of the solvent and the resulting risk of workers’ exposure.
Point 8. Page 2, line 86: The authors claim that GNPs “represent a good candidate to replace carbon nanotubes for many applications”. However, this applies also to other conducting carbon nanomaterials including graphene itself, carbon nanofibers,… Please rewrite this sentence.
Response 8: In order to complete the sentence, we rewrote it as follows (page 2 lines 86-89):
1) “Therefore, these nanostructures represent a good candidate to replace many carbon-based conductive nanomaterials such as carbon nanotubes, carbon nanofibers, carbon dots and fullerenes, for many applications, due to the low manufacturing cost and easy production scalability.”
Point 9. The graphitic structures shown in Figure 1 (b) correspond to worm-like expanded graphite, not “warm-like” expanded graphite. Also, I do not think the term nanomaterials should be capitalized, and it does not require here the use of the acronym “NMs” (page 1, line 44).
Response 9: Thank you for the suggestion, we replaced the term “warm” with “worm” and we wrote “nanomaterials” instead of the acronym “NMs”.
Point 10. Correct typos and language.
Response 10: We have corrected some typos and the language.

Reviewer 4 Report
The rapid development of the interdisciplinary field of nanotechnology, integrating the knowledge in physics, chemistry and biology causes the issue of occupational safety related to nanotechnological processes to emerge. This important problem is addressed in the present paper, which describes the experimental study of workers’ exposure during the relevant technological processes. The processed are aimed at production of graphene nanoplatelets in two steps: thermal expansion and liquid exfoliation. In both steps, performed in separate laboratories, the exposure of the involved workers to graphene nanoplatelets is assessed using a multiinstrumental approach. The exposure is measured using the Condensation Particle Counter, Mini Diffusion Size Classifier, Personal impactor supplemented with analysis of the morphology and chemical composition of nanoparticles using Field Emission Scanning Electron Microscope with Energy Dispersive X-ray Spectroscopy. The stages of the whole process which are connected with the most pronounced exposure are identified and the exposure is assesses in quantitative way.
The paper presents interesting results of importance to the occupational safety and health related to modern nanotechnological processes. It is careful, scientifically sound and of interest to the Readers. I recommend the manuscript for publication in Nanomaterials journal, after the Authors give consideration to the points listed below:
- Line 127: Maybe it could be useful to mention what sort of filters was used in the mentioned masks?
- Line 185: Maybe it would be useful to emphasize that the laboratory Lab3 is identical to Lab1 (if it is true) – this could be more clear.
- Line 190, line 203, lines 421-422: Some sort of compilation error is probably producing the communicates in bold – I guess some reference to table should be placed there instead of the communicates.
- Lines 102 and 109: is the unit “gr” denoting grain? Maybe it could be useful to convert it to SI units or just mention the relevant value in SI units in the bracket?
- Line 605: “Instrument”
- Figure 1A, panel (b): “LINEAR”
- Figures 1-4: a general remark: I guess it could be highly useful to add the confidence intervals to the linear regression lines in all the plots, if possible.
- Tables 1A, 2A: a general remark: it would be useful to add the confidence intervals for the fitted regression parameters α and β
Author Response
Authors thank the reviewer for his valuable and constructive comments and suggestions. We have carefully analyzed them and consequently we have improved some parts of the manuscript. Changes are marked in yellow in the revised manuscript. In the following, the list of changes and answers to the reviewer’s comments are reported.
Point 1. Line 127: Maybe it could be useful to mention what sort of filters was used in the mentioned masks?
Response 1: We adjusted the sentence specifying the filters used in the masks (page 1 lines 137-139).
“During these two operations all the workers wear protective gloves, laboratory coats and full-face respirators (mod. 3M™ 6000 series) equipped with EPA filters (mod. 3M™ 6099 ABEK2 P3 series).”
Point 2. Line 185: Maybe it would be useful to emphasize that the laboratory Lab3 is identical to Lab1 (if it is true) – this could be more clear.
Response 2: We adjusted the sentence (page 4 lines 197-200) in order to emphasize that the laboratory 3 is identical to Lab1.
“Another laboratory (lab 3), in which no nanomaterials were produced, was selected for the simultaneous bkgFF measurements, by using DM-UF4. In particular, lab 3, located next to the lab 2, has same orientation, structural and ventilation properties (both natural and mechanical) as the lab 1 and 2.”
Point 3. Line 190, line 203, lines 421-422: Some sort of compilation error is probably producing the communicates in bold – I guess some reference to table should be placed there instead of the communicates.
Response 3: We are sorry for the compilation error; We adjusted the references linked to the tables.
Point 4. Lines 102 and 109: is the unit “gr” denoting grain? Maybe it could be useful to convert it to SI units or just mention the relevant value in SI units in the bracket?
Response 4: We made a mistake; we would like to indicate the unit “grams”. We have changed the abbreviation “gr” in “g”.
Point 5. Line 605: “Instrument”
Response 5: We have corrected the typos in line 642.
Point 6. Figure 1A, panel (b): “LINEAR”
Response 6: We have corrected the typos in the graph.
Point 7. Figures 1-4: a general remark: I guess it could be highly useful to add the confidence intervals to the linear regression lines in all the plots, if possible.
Response 7: Since we used all real-time devices placed in different sampling points for simultaneous measurements, at the beginning and at the end of each campaign instrument comparison sessions were conducted at the same sampling point, in order to define the correlations among the instruments measuring the same parameter and to align the obtained values. Correlations have been observed in the same operating conditions in each workplace, as reported also in (Asbach et al 2012). R2 and Pearson’s correlation coefficient (Pearson’s R) have been used to test each correlation, after outlier’s removal. For this reason, we didn’t feel it was necessary to deeply analyze the linear regression, by calculating the confidence intervals, for the purposes of our study.
In any case, we reported in the graphs the standard error of the intercept (β) and the slope (α).
[Asbach et al 2012]: Asbach C, Kaminski H, von Barany D, et al. Comparability of portable nanoparticle exposure monitors. Ann Occup Hyg. 2012;56(5):606-621. doi:10.1093/annhyg/mes033.
Point 8. Tables 1A, 2A: a general remark: it would be useful to add the confidence intervals for the fitted regression parameters α and β
Response 8: Please, see the answer in point 7.

Round 2
Reviewer 2 Report
Thank the author for looking and replying into my comments.
All of the following are just my suggestions for future works (not for the current manuscript).
If cost and condition permit, may be more studies can be done to see the variations of the measurement data conducted at the same exfoliation temperature, with all other conditions unchanged (that is in air-conditioned enclosed environment with same window opened at the thermal expansion phase). Moreover, additional studies can be done to investigate on changing each of the factor on the variation of the measurement data. For example, changing the exfoliation temperature, operated by different workers, different room pressure and opening/closing different windows/doors. Otherwise, the reader will have difficulty to be convinced if the variation is due to changing temperature, or the inherent process (that is some fluctuation in measurement data for the same procedure, like in your current script). We have to know the repeatability of a particular experiment procedure before moving onto changing each variable one by one within the procedure for comparison.
For my previous comment, “In materials and methods section, the author may consider revising the scanning electron microscope images of graphite intercalated compounds in the result section.”, this sentence means that the current SEM images (Fig. 1 in the material and methods), should be considered as results and to be moved to the result section. Moreover, in the materials and methods section should just give the information on the equipment used, e.g., atomic force microscopy (Veeco ICON), and scanning electron microscope (JOEL JSM-7401F). This style is only my preference, may be clearer to the reader.
For the conclusion, it is fine to keep the conclusion in this form. To make the conclusion (and abstract) to be more conclusive and precise is just my preference because (seems to me that) this style will help the reader to have better a understanding of the key novelty/idea of your work, to improve citation and acceptance rate of your manuscript.
Author Response
Authors thank the reviewer for his observations and suggestions. In the following, the answers to reviewer’s comments is reported.
All of the following are just my suggestions for future works (not for the current manuscript):
Point 1: If cost and condition permit, may be more studies can be done to see the variations of the measurement data conducted at the same exfoliation temperature, with all other conditions unchanged (that is in air-conditioned enclosed environment with same window opened at the thermal expansion phase). Moreover, additional studies can be done to investigate on changing each of the factor on the variation of the measurement data. For example, changing the exfoliation temperature, operated by different workers, different room pressure and opening/closing different windows/doors. Otherwise, the reader will have difficulty to be convinced if the variation is due to changing temperature, or the inherent process (that is some fluctuation in measurement data for the same procedure, like in your current script). We have to know the repeatability of a particular experiment procedure before moving onto changing each variable one by one within the procedure for comparison.
Response 1: We are agreed with the reviewer because the different environmental conditions could influence the measurements. So effectively, we are planning to repeat the experiments with the same two temperatures and the same conditions in order to verify the repeatability of the procedure. For future research work, we would like to conduct the same two experimental campaigns by changing some important factors that can influence the measurements, such as keeping the windows and doors closed or open or the ventilation system turned on/off.
Point 2: For my previous comment, “In materials and methods section, the author may consider revising the scanning electron microscope images of graphite intercalated compounds in the result section.”, this sentence means that the current SEM images (Fig. 1 in the material and methods), should be considered as results and to be moved to the result section. Moreover, in the materials and methods section should just give the information on the equipment used, e.g., atomic force microscopy (Veeco ICON), and scanning electron microscope (JOEL JSM-7401F). This style is only my preference, may be clearer to the reader.
Response 2: Thank you for your suggestions. However, since in the “materials and methods” section we reported the general description of the nanomaterials involved in the production process as preliminary investigation about their morphological features, we included the SEM characterization of the produced materials in that paragraph. We will certainly consider your suggestion for our future works.
Point 3: For the conclusion, it is fine to keep the conclusion in this form. To make the conclusion (and abstract) to be more conclusive and precise is just my preference because (seems to me that) this style will help the reader to have better a understanding of the key novelty/idea of your work, to improve citation and acceptance rate of your manuscript.
Response 3: Thank you very much for the suggestion. In our future works we will try to make the conclusion section more concise.
